# Optical control of polarization in ferroelectric heterostructures

Tao Li[1], Alexey Lipatov [2], Haidong Lu[1], Hyungwoo Lee[3], Jung-Woo Lee[3], Engin Torun[4], Ludger Wirtz[4], Chang-Beom Eom[3], Jorge Íñiguez[5], Alexander Sinitskii [2] & Alexei Gruverman [1]

In the ferroelectric devices, polarization control is usually accomplished by application of an electric field. In this paper, we demonstrate optically induced polarization switching in $BaTiO_3$-based ferroelectric heterostructures utilizing a two-dimensional narrow-gap semiconductor $MoS_2$ as a top electrode. This effect is attributed to the redistribution of the photo-generated carriers and screening charges at the $MoS_2/BaTiO_3$ interface. Specifically, a two-step process, which involves formation of intra-layer excitons during light absorption followed by their decay into inter-layer excitons, results in the positive charge accumulation at the interface forcing the polarization reversal from the upward to the downward direction. Theoretical modeling of the $MoS_2$ optical absorption spectra with and without the applied electric field provides quantitative support for the proposed mechanism. It is suggested that the discovered effect is of general nature and should be observable in any heterostructure comprising a ferroelectric and a narrow gap semiconductor.

[1] Department of Physics and Astronomy, University of Nebraska, Lincoln, NE 68588, USA. [2] Department of Chemistry, University of Nebraska, Lincoln, NE 68588, USA. [3] Department of Materials Science and Engineering, University of Wisconsin, Madison, WI 53706, USA. [4] Physics and Materials Science Research Unit, University of Luxembourg, L-1511 Luxembourg, Luxembourg. [5] Department of Materials Research and Technology, Luxembourg Institute of Science and Technology, 5 Avenue des Hauts-Fourneaux, L-4362 Esch/Alzette, Luxembourg. Correspondence and requests for materials should be addressed to A.G. (email: agruverman2@unl.edu)

The characteristic feature of the ferroelectric materials is the presence of the reversible spontaneous polarization that can be switched by an electric field[1]. Switchability of ferroelectric polarization enables control of a number of polarization-dependent electronic, mechanical, optical, and other functional properties, which forms the basis of their device applications[2]. The most recently reported effects of this nature include ferroelectrically induced resistive switching phenomena and the associated memristive behavior[3], electrical control of antiferromagnetic domains[4], modulation of the electronic transport in 2D semiconductors[5–7] and phase transitions at magnetic complex oxide interfaces[8,9]. Although polarization reversal is typically realized via application of an electric field, recently it has been shown that mechanical stress and chemical environment can also be used as external stimuli for polarization control[10–12].

Among the most important properties of ferroelectrics is their strong interaction with light, which gives rise to a variety of the photo-induced phenomena coupled to polarization. This includes experimentally observed effects, such as photovoltaic behavior[13] and photostriction[14,15], as well as expected but not demonstrated features, such as optically generated interface metal-insulator transitions[16]. Other reported optically enabled effects include UV-induced domain pinning[17,18], increased imprint[19,20], domain wall displacement by varying the polarization angle of a laser[21], THz radiation via optically induced modulation of polarization[22]. These observations provide tangible reasons for investigation of the interaction between light and ferroic order parameters and for development of nanostructures with the physical functionality controlled by light.

In this paper, we report optically induced switching of polarization in the hybrid $MoS_2/BaTiO_3/SrRuO_3$ tunnel junctions[23] realized via photo-absorption in two-dimensional $MoS_2$—a transition metal dichalcogenide semiconductor, characterized by a strong and fast photoresponse. Monolayer $MoS_2$ has a direct optical gap of ~1.9 eV, while bulk $MoS_2$ shows indirect bandgap of ~1.2 eV[24–26]. Further, taking advantage of the $MoS_2$ photosensitivity, we demonstrate that optical excitation also leads to a sizable change in the perpendicular-to-plane electronic transport

in the $MoS_2/BaTiO_3/SrRuO_3$ tunnel junctions—an effect that we term as optical electroresistance effect (OER). The observed effects may open possibilities for remote control of the electronic properties of ferroelectric-based devices for advanced optoelectronic applications. It should be noted, however, that for any future applications it is important to find out if the speed of optical switching could achieve the same scale as polarization reversal induced by electrical means.

## Results

**Electrical control of polarization and resistive switching.** For this study, we employ hybrid ferroelectric tunnel junctions (FTJs) comprised of epitaxial ferroelectric $BaTiO_3$ (BTO) film with the thickness ranging from 6 to 12 unit cells (u.c.) (i.e., from 2.4 nm to 4.8 nm) sandwiched between the top $MoS_2$ and bottom $SrRuO_3$ (SRO) electrodes. Details of the growth of the BTO/SRO heterostructures on the (001) $SrTiO_3$ substrates by pulsed laser deposition (PLD)[27] are given in Methods section. Multilayer $MoS_2$ flakes were transferred from the $MoS_2$ single crystal to the BTO film surface via mechanical exfoliation method using an adhesive tape[28] (see Methods for details). The polarization state of the BTO and changes in the transport properties of the $MoS_2$/BTO/SRO tunnel junctions induced by the electrical and/or optical means have been probed by piezoresponse force microscopy (PFM) and conducting atomic force microscopy (C-AFM), respectively.

Figure 1a–d shows the results of the electrical polarization switching in the $MoS_2/BTO(12\ u.c.)/SRO$ junctions. Preliminary studies showed that the fabricated BTO films had the polarization aligned in the direction perpendicular to the surface, and therefore only 180° inversion of polarization was allowed (see Supplementary Figure 1). Application of the electrical bias to the $MoS_2$ flake leads to the reversal of polarization in the BTO film underneath as evidenced by the contrast inversion in the PFM phase images (Fig. 1a, c). A reduced PFM amplitude signal for the upward BTO polarization state (Fig. 1b, d), consistent with our previously reported results[29], is due to the change in the $MoS_2$

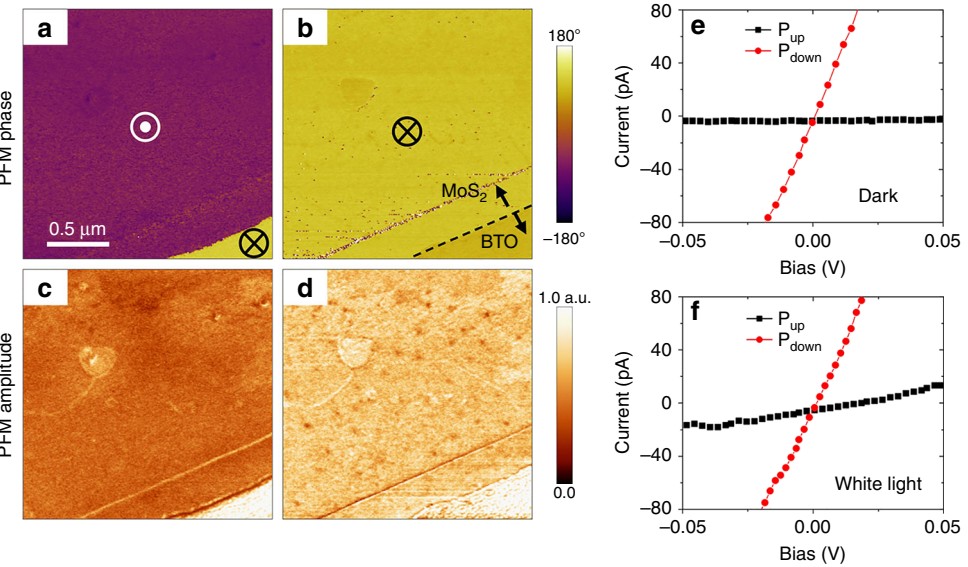

**Fig. 1** Electrically induced polarization switching in the $MoS_2/BaTiO_3/SrRuO_3$ junction. **a**, **b** PFM phase (**a**) and amplitude (**b**) images after application of a negative voltage pulse (−5 V, 0.5 s) to the $MoS_2$ flake. The 12-u.c.-thick BTO film underneath the $MoS_2$ flake is fully switched to the upward polarization, $P_{up}$. **c**, **d** PFM phase (**c**) and amplitude (**d**) images after application of several positive voltage pulses (+5 V, 0.5 s) to the $MoS_2$ flake. BTO underneath the $MoS_2$ flake is fully switched to downward polarization, $P_{down}$. The polarization state of the bare BTO film (at the lower right corner) is not affected by the electrical bias. **e**, **f** The *I–V* characteristics of the same junction measured in the dark and during illumination. The tunneling current for the OFF state ($P_{up}$) is largely increased under illumination

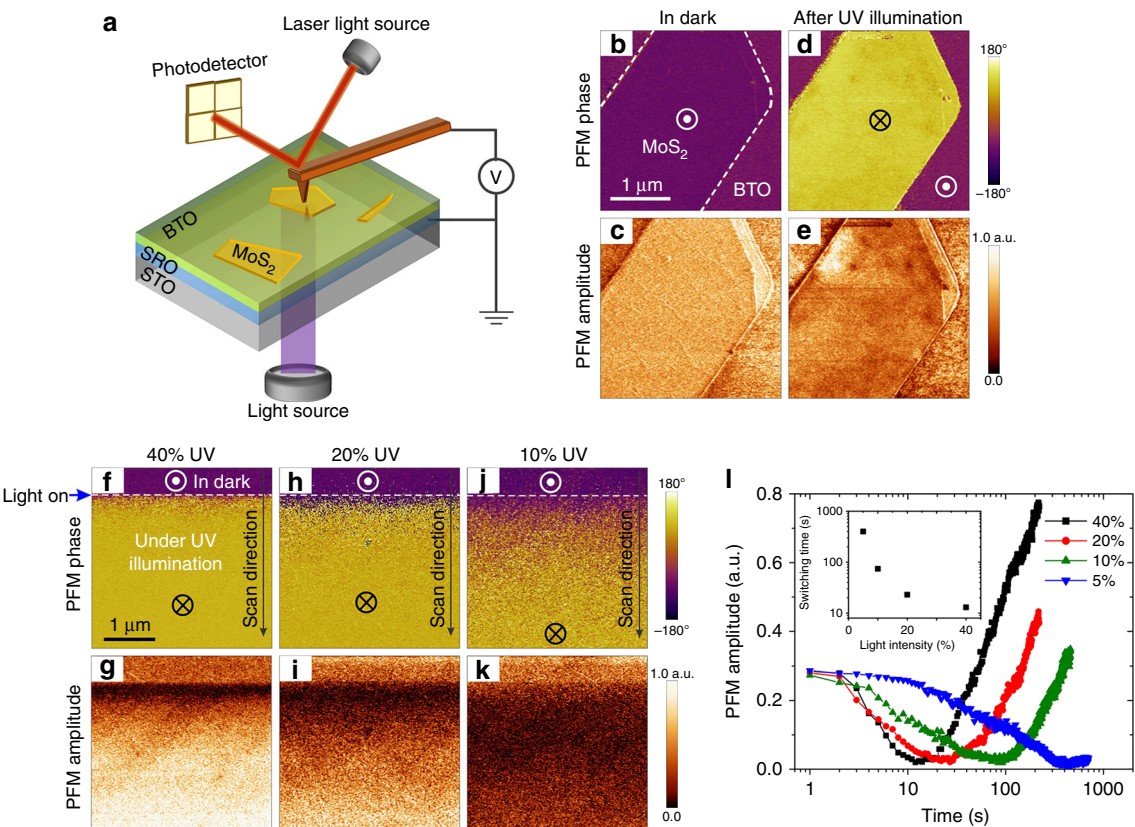

**Fig. 2** Optically induced changes of the polarization in $MoS_2$/$BaTiO_3$/$SrRuO_3$ junctions. **a** A sketch of the experiment geometry. **b–e** PFM phase (**b**, **d**) and amplitude (**c**, **e**) images acquired in the dark before and after UV illumination. The $MoS_2$ flake boundary is indicated by the dashed lines in **b**. The BTO film underneath $MoS_2$ was electrically poled to the upward polarization before illumination. **f–k** PFM phase (**f**, **h**, **j**) and amplitude (**g**, **i**, **k**) images under illumination with light of different intensity: **f**, **g** 40, **h**, **i** 20, and **j**, **k** 10% of the nominal light source power. The scanned area is fully covered by $MoS_2$. Polarization was electrically switched in the dark to the upward polarization before each scan. **l** PFM amplitude signal as a function of time under illumination based on the analysis of the PFM images in **g**, **i**, **k** (PFM images for 5% light intensity are not shown). The minimum of PFM amplitude signal indicates an unpolarized state (likely due to the equal fraction of the upward and downward domains). The inset shows the optical switching time (defined as a time required for the PFM amplitude to reach its minimum value after turning on the light) as a function of light intensity

conductivity. Earlier, it has been shown that $MoS_2$ in the $MoS_2$/BTO/SRO junctions behaves as a good conductor when the BTO polarization is pointing downward and tends to be more insulating for the opposite polarization. Both polarization states are found to be stable for at least several hours without any significant decay. The I–V characteristics in Fig. 1e measured in the dark illustrate the resistive switching effect in the $MoS_2$/BTO/SRO junctions associated with the polarization reversal in the BTO barrier and the changed conductivity of $MoS_2$. Specifically, the downward BTO polarization, $P_{down}$, corresponds to the low resistance (ON) state and the upward polarization, $P_{up}$, produces the high resistance (OFF) state, typically yielding an OFF/ON ratio of the order from several hundred to several thousand[29].

**Optical electroresistance effect**. Due to the photosensitivity of $MoS_2$ it is reasonable to expect that optical illumination should lead to the appearance of an electric field due to photogenerated charge carriers that would affect the electronic properties of the $MoS_2$/BTO/SRO junctions[30]. The I–V testing of the $MoS_2$/BTO/SRO junctions after white light illumination for <10 min (Fig. 1f) reveals that resistance of the OFF ($P_{up}$) state decreases from 70 to 3 GΩ upon optical excitation, but stays almost intact for the ON ($P_{down}$) state (~200 MΩ), leading to the reduction of the OFF/ON ratio from 350 (measured in the dark) to 60. This effect is similar to the optically induced changes in the transport properties observed in the $MoS_2$-gated $Pb(Zr,Ti)O_3$ (PZT) field effect

transistors[7]. It was also found that optical illumination of the $MoS_2$-PZT structure causes an abrupt decrease in the PFM amplitude signal. To explain this behavior, two hypotheses were proposed. The first one attributed it to domain rearrangement, i.e., a change in the polarization state, due to the less efficient polarization screening and the second one invoked optically induced changes in the resistance of $MoS_2$.

**Optically induced polarization reversal**. To clarify the interplay between the BTO polarization state, $MoS_2$ resistance and optical excitation, further studies of the $MoS_2$/BTO(12 u.c.)/SRO junctions have been carried out by acquiring the PFM images and monitoring the changes in the PFM signal under constant UV illumination (with a center peak of 377 nm corresponding to 3.29 eV). Geometry of the experimental setup is shown in Fig. 2a. The light intensity measured at the BTO surface is about 5 mW cm$^{-2}$ for 40% UV light output power. Figure 2b–e illustrates the effect of optical illumination on the PFM images of the $MoS_2$/BTO/SRO junction, which was initially poled to the upward polarization state by application of the negative electrical bias to $MoS_2$ in the dark (Fig. 2b, c). It can be seen that after illumination the BTO film covered with $MoS_2$ exhibits an inversed PFM phase contrast while the PFM signal of the bare BTO films shows almost no change (Fig. 2d, e). This behavior resembles the $MoS_2$/BTO response to the electrical poling by a positive bias and, thus, can be attributed to the optically induced polarization reversal. Note,

that this effect is observed only in the $MoS_2$/BTO/SRO junctions in the upward polarization state and no optically induced changes have been detected in the samples with the downward polarization. To get reference data, bare BTO films with electrically poled upward and downward domains were illuminated using the same UV light intensity. In this case, the PFM amplitude signal was changing due to the effect of the photogenerated carriers in the BTO film. However, no polarization switching was observed (see Supplementary Figure 2) although redistribution of the photo-induced charges under UV illumination was detected by Kelvin Probe Force Microscopy (KPFM). Thus, it is important to emphasize that the optically induced switching occurs only in the $MoS_2$/BTO/SRO junctions and does not occur in the BTO/SRO structures. As will be shown below, this effect can be explained by the interplay between the redistribution of the photogenerated charges in $MoS_2$ and BTO. In addition, to check if the same effect would be induced by the light of a different wavelength, we have performed illumination by green (562 nm) light. It has been found that no optically induced polarization switching was observed for the same light intensity and for the comparable time of illumination.

It has been also found that the rate, with which the PFM signal responds to the optical excitation, depends on the light intensity: the response becomes slower upon a light power decrease. This effect is illustrated in Fig. 2f–k where the scanning of the sample, initially poled to the upward polarization, begins from the top of the image in the dark and then the light is turned on at the moment indicated by a blue arrow. This leads to an abrupt reduction of the PFM amplitude signal and fuzzy PFM phase contrast. However, as scanning under illumination continues, the PFM phase signal undergoes complete inversion and, for the light source at 40% of its nominal power, within several seconds the PFM amplitude completely recovers (Fig. 2f, g). It takes a much longer time for the PFM amplitude to recover if the light source power is reduced to 20 or 10% (Fig. 2h–k) and at 5% of the nominal source power the reduced PFM amplitude signal does not recover irrespective of how long the sample is exposed to light. The time dependent behavior of the PFM amplitude signal

is shown in Fig. 2l, where the inset illustrates the effect of the light intensity on the time, at which the PFM amplitude reaches its minimum value before starting to increase—a parameter provisionally termed as an optical switching time. To rule out the artifacts caused by the AFM detection system, the PFM images of the bare BTO surface were used as a reference. It has been confirmed that the described changes in the amplitude and phase signals are confined to the region covered by the $MoS_2$ flake.

Taking all these observations together, we argue that the changes in the PFM response shown in Fig. 2 are a result of the optically induced polarization reversal caused by the photogenerated charge carriers in $MoS_2$. The abrupt decrease in the PFM amplitude upon illumination and its subsequent increase is reminiscent of the PFM visualization of the polarization reversal induced by electrical or mechanical means[11,31], which presumably proceeds via nucleation and growth of ultrasmall domains of opposite polarity. To confirm that the BTO underneath $MoS_2$ has been indeed switched by the UV light, we have removed part of the $MoS_2$ flake after illumination and performed PFM imaging of the exposed BTO surface. The obtained PFM maps reveal the downward polarization state in the exposed BTO (see Supplementary Figure 3), which is a solid evidence of the optically induced polarization switching in the $MoS_2$/BTO/SRO junction ruling out any artificial imaging effect. In addition, to exclude the heating effect due to light absorption as a possible reason for the observed switching, the junction has been heated up to 100 °C in the dark. In this case, no polarization switching was observed neither underneath the $MoS_2$ nor in the bare BTO film (see Supplementary Figure 4).

**Surface potential and tunneling current measurements**. To further verify the effect of optically induced switching, we have measured the surface potential of the $MoS_2$/BTO/SRO junctions before and after UV illumination by KPFM (Fig. 3). Figure 3a shows $MoS_2$ flakes with the BTO underneath electrically poled to the upward and downward polarization states (flakes M1 and M2,

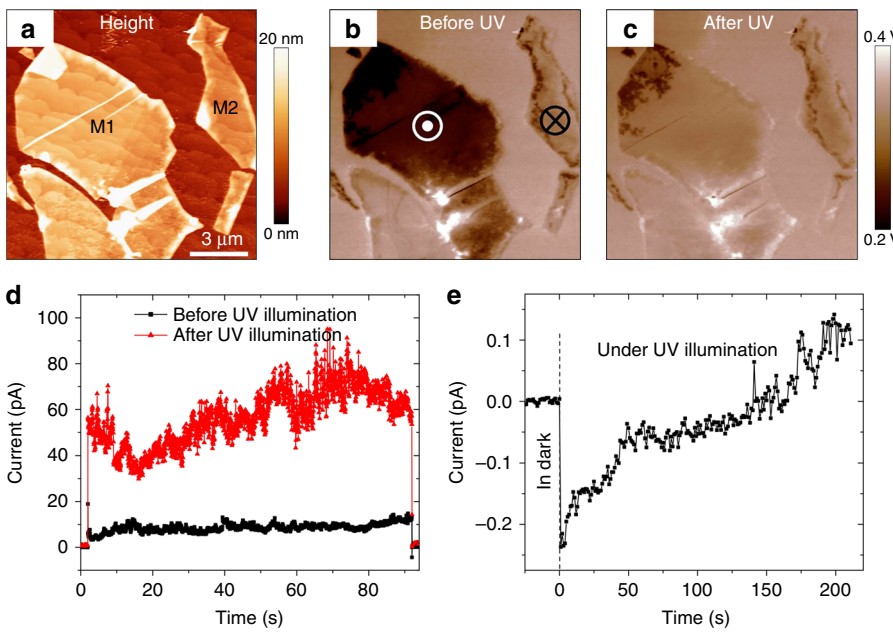

**Fig. 3** Optically induced change in surface potential and tunneling current. **a** AFM topography of the $MoS_2$ flakes on the BTO surface. **b** A surface potential map obtained in the dark after electric poling of BTO under the $MoS_2$ flakes (under M1 flake – upward, and under M2 flake – downward). **c** A surface potential map of the same flakes measured in the dark after optical illumination. **d** Tunneling current measured in the dark before and after illumination, with a DC bias of 0.5 V. **e** A zero-bias photocurrent measured in the dark and under illumination. In **d** and **e**, the initial polarization state of the $MoS_2$/BTO/SRO junction was upward

respectively). The corresponding PFM images are shown in Supplementary Figure 5. The KPFM measurements performed in the dark before illumination reveal that the MoS$_2$/BTO junction with the downward BTO polarization (M2) exhibits a higher potential than the junction with the upward polarization of BTO (M1) (Fig. 3b) in agreement with our previous results[29]. After UV illumination, the surface potential of the M1 flakes became more positive (Fig. 3c), while the surface potential of M2 did not change. The apparent increase of the surface potential of M1 upon illumination is consistent with the optically induced switching of polarization from the upward to downward state.

An additional confirmation of the photo-induced polarization reversal effect is a gradual change of the photocurrent magnitude and direction under illumination (Fig. 3e). Also, tunneling current measured for the M1 flake, where BTO was initially polarized upward, showed a significant increase after illumination for 20 min (Fig. 3d), which is consistent with the larger tunneling current for the downward state (Fig. 1e).

We have evaluated the reproducibility of the optically induced switching of the MoS$_2$/BTO/SRO junction by repeatedly switching it to the upward polarization by the electrical bias applied in the dark, and then subjecting it to UV illumination at the 40% intensity level. The junctions are still optically switchable after the cumulative illumination time of about 4 h. However, with each cycle, the optical switching time to the downward polarization is gradually increasing from tens to hundreds of seconds. The photo-oxidation and structural degradation of MoS$_2$ under prolonged optical illumination has been reported earlier[32]. Thus, it can be assumed that an increase in the switching time is a result of the optically induced degradation of the MoS$_2$ electronic properties.

## Discussion

Below, we propose a physical mechanism of the photo-induced polarization switching in the MoS$_2$/BTO/SRO tunnel junctions. The main premise of proposed model is based on the fact that in spite of the polarization-dependent redistribution of the photo-generated carriers, the UV light does not induce polarization reversal in BTO—this effect is observed only in the presence of the MoS$_2$ layer on top of BTO. It should be also mentioned that in the FTJs capped with metallic electrode, no photo-induced polarization reversal has been observed either[33]. Hence, we have to consider the intrinsic effects associated with MoS$_2$ to understand the mechanism of the optically induced polarization reversal MoS$_2$/BTO/SRO tunnel junctions. Next, it is important to note that the FTJs are essentially asymmetric structures characterized by a strong imprint and a preference of one polarization state over the opposite one. Specifically, the MoS$_2$/BTO/SRO tunnel junctions exhibit a negative imprint with $P_{down}$ being the preferred polarization state, which is equivalent to the presence of the built-in field $E_{bi}$ oriented toward the bottom electrode (Fig. 4a). Yet, in spite of this preference for the $P_{down}$ state, it is possible to switch the polarization to the $P_{up}$ state by applying a negative bias to the probing tip. Injection of electrons and their accumulation at the MoS$_2$/BTO interface (charge σ < 0 in Fig. 4) screens the upward polarization and leads to the appearance of a downward oriented electric field $E'_\sigma$ in the MoS$_2$ flake (Fig. 4a). At the same time, in BTO, these electrons generate an electric field $E_\sigma$ pointing upward. The $E'_\sigma$ field will change the MoS$_2$ electronic properties, so that under optical illumination it will facilitate the formation of inter-layer excitons with a dipole direction perpendicular to the interface (Fig. 4b). Extended exposure to light leads to an increased accumulation of the photo-induced charges at the interface (so that σ~0), destabilization of the $P_{up}$

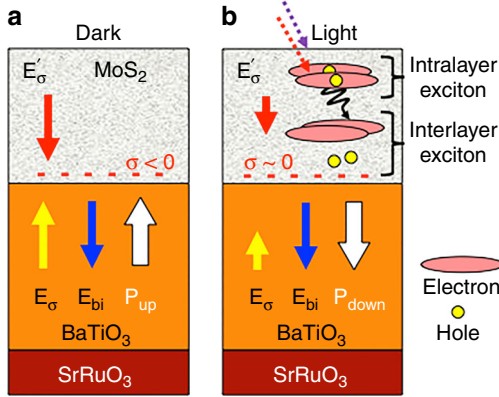

**Fig. 4** Mechanism of the photo-induced switching. **a** Illustration of the electric fields $E_\sigma$ and $E'_\sigma$ in BTO and MoS$_2$, respectively, a built-in electric field $E_{bi}$ and the negative charge σ accumulated at the interface when BTO is switched to the $P_{up}$ state. **b** Generation of the polar excitons under optical illumination resulting in compensation of the interfacial charge σ, decrease of the associated field $E_\sigma$ and polarization reversal to the $P_{down}$ state favored by $E_{bi}$

state via decrease of the $E_\sigma$ field and eventual switching of polarization to the downward direction.

To test this conjecture, we have calculated the electronic and optical properties of the MoS$_2$ tri-layers (Fig. 5a) using density functional theory (DFT) and many-body perturbation theory[34] (see Methods and Supplementary Note 1 for details). The chemical interfacial effects have been ignored in these simulations; yet, we do capture the main consequences of charge accumulation at the MoS$_2$/BaTiO$_3$ interface by modeling the presence of the corresponding electric field $\left(E'_\sigma\right)$ by means of a saw tooth potential. Representative results are shown in Fig. 5b–f). The calculated band structure (Fig. 5b, c) shows an indirect gap character, but the optical properties at low excitation energies are determined by the excitonic states formed from electron–hole pairs around the direct gap at the high-symmetry point K[35]. Without the electric field, the orbitals corresponding to the valence band maximum (VBM) form bonding and anti-bonding orbitals due to the interaction between the MoS$_2$ monolayers. This results in a splitting of the VBM into several sub-bands (Fig. 5b). The role of the polarization is manifested by the effect of the electric field produced by the interfacial charge σ on the properties of MoS$_2$. We have calculated the MoS$_2$ band structure in the presence of the a perpendicular homogeneous electric field of 0.1 V Å$^{-1}$, which is equivalent to σ of about 0.07 electrons per BTO unit cell at the MoS$_2$/BTO interface. In this case, the onsite energies on the layers change, so that the highest VBM is mostly localized on layer 1 and the lowest one on layer 3 (Fig. 5c). Similarly, the conduction band minimum (CBM) is split into bands with orbitals localized on one layer. Here, the lowest independent-particle excitation energy corresponds to an electron in the CBM on layer 3 and a hole in the VBM on layer 1. The underlying effect can be seen as the result of a Stark effect, shifting the conductance and valence band orbitals on opposite sides in opposite directions (see zoom into the region around K in Fig. 5c). The result is a "staggered" band gap, similar to what is observed in hetero-bilayers of transition metal dichalcogenides[36].

Note that a careful consideration of the optical properties of MoS$_2$ must take into account the electron–hole attraction, which can lead to excitonic effects. This excitonic binding is quite strong (up to 0.5 eV) if electrons and holes are in the same layer ("intra-layer exciton")[35] but can also be present if electrons and holes are on different layers ("inter-layer exciton"). In the absence of electric field, the intra-layer exciton is the lowest optical

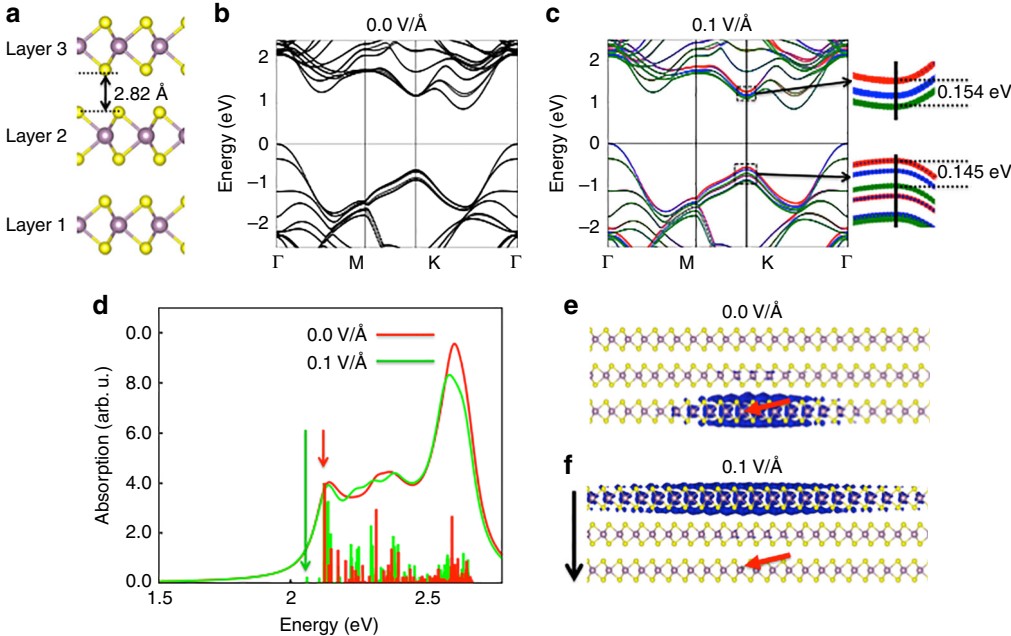

**Fig. 5** Modeling of the electronic and optical properties of $MoS_2$. **a** Optimized atomic structure of trilayer $MoS_2$. The purple and the yellow balls are Mo and S atoms, respectively. **b**, **c** Band structure of trilayer $MoS_2$ without (**b**) and with (**c**) an electric field. In **c**, the different colors represent the bands projected onto the three correspondingly labeled layers in **a**. A separation of the valence and conduction bands as well as the energy difference between them at K are shown as a zoom-in inset in **c**. **d** Calculated absorption spectra with (green) and without (red) an electric field. Positions of the relevant intra- and inter-layer excitonic peaks are marked with vertical lines. **e**, **f** Excitonic wave functions of the first excitonic peak (intra-layer exciton) in the absorption spectrum without the electric field (**e**) and of the first excitonic peak (inter-layer exciton) in the presence of the electric field (**f**). The direction of the electric field is shown with a vertical black arrow. In **e** and **f**, the electron density is plotted for the case when the hole (marked by a small red arrow) is localized in layer 1 close to Mo atoms

excitation. However, at a sufficiently strong field, the energy splitting of VBM and CBM, and thus the reduction of the band gap, get large enough, such that an inter-layer exciton becomes the lowest optical excitation. Earlier, a similar conclusion was reached for bilayer $MoS_2$[37]. This scenario is quantitatively verified by our calculations of the optical absorption spectra with and without the electric field (Fig. 5d). It can be seen that although the low-energy spectrum (in the range from 1.5 eV to 3.0 eV) is dominated by the intra-layer exciton, for the $0.1$ V Å$^{-1}$ field, the lowest peak in the absorption spectrum corresponds to an inter-layer exciton. Note that the probability for absorption into this excitonic peak is vanishingly small. It can be suggested that the most probable scenario of the inter-layer excitons formation is through the optical absorption into an intra-layer excitons with subsequent non-radiative relaxation to the inter-layer excitonic state with the lower energy. As shown in Fig. 5e, f, formation of the inter-layer excitons involves a separation of the electron–hole pairs along the direction of the applied field. In the case of the field in the $MoS_2$ layer pointing toward the $MoS_2/BaTiO_3$ interface, the corresponding low-energy polar excitons will move holes from the upper $MoS_2$ layers towards the interface with $BaTiO_3$ where the hole concentration will increase with time of light exposure.

With all the experimental and modeling data in mind, the optically induced switching in the $MoS_2/BTO/STO$ junctions can be explained as a result of the interplay between the photo-generated charges in $MoS_2$ and polarization charges in BTO. Light absorption in the $MoS_2$ electrode occurs via the dominant intra-layer excitons, which eventually decay into inter-layer excitons, as sketched in Fig. 4. This two-step process results in the long-lived electronic states that tend to bring positive carriers to the $MoS_2/BaTiO_3$ interface thereby compensating the electronic charge σ, which screens the upward polarization. Thus,

illumination reduces the $E_\sigma$ field acting on $BaTiO_3$ and facilitates the reversal of polarization to the preferred downward $P_{down}$ state under the action of the built-in field $E_{bi}$. Note also that, according to this physical picture and our experimental evidence, light cannot destabilize the $P_{down}$ state of the $MoS_2/BTO/SRO$ junction. In the $P_{down}$ state we have σ~0 at the interface, and $E_\sigma \sim 0$ inside the $MoS_2$ flake. In this case, there are no low-lying dipolar excitons in $MoS_2$, illumination will not result in charges moving to the interface, and no switching to $P_{up}$ will occur. As a final note, careful investigation of the wavelength dependence of the optical switching behavior is necessary to get a better understanding of the role of the photo-induced charge redistribution in BTO in the observed effect.

To emphasize the general nature of this phenomenon, it is important to note that the observed optical electroresistance effect is not contingent on any specific property of the ferroelectric barrier. This means that it should be present in any hybrid tunnel junction comprised of a ferroelectric and photo-absorbing narrow-band semiconductor. To prove this point, we performed similar measurements by illuminating the $WSe_2/BTO/STO$ junctions employing two-dimensional semiconductor $WSe_2$ as a top electrode. The optical switching results resembling the behavior observed in the $MoS_2/BTO/STO$ junctions are shown in Supplementary Figure 6. This confirms our conclusion that accumulation of a sufficiently high concentration of photo-generated charges at the semiconductor/ferroelectric interface should result in optically induced polarization reversal and associated changes in the transport properties that can be exploited in opto-ferroic devices with light-triggered functionalities. The first-principles simulations suggesting that the switch is made possible by inter-layer polar excitons indicate a potentially strong dependence of the observed effect on the thickness of the 2D layer. An important challenge lying ahead is a realization of

the bi-directional optical polarization switching. This can be potentially achieved by employing ambipolar 2D semiconductors, which could change their photo-induced electronic properties to favor either polarization direction. A serious question that needs to be addressed is how fast the polarization can be switched by optical means.

## Methods

**Sample preparation**. Epitaxial BTO thin films were grown on the SRO layers deposited on single-crystalline (001) STO substrates by pulsed laser deposition (PLD). The epitaxial growth of the films was monitored by in-situ high-pressure reflection high-energy electron diffraction (RHEED). Before the film growth, the STO substrates were etched by buffered-HF for 1 min and annealed at 1000 °C for 6 h to obtain atomically smooth and $TiO_2$-terminated surfaces. During the film growth, the temperature of substrates and the oxygen partial pressure were kept at 610 °C and 0.12 mbar, respectively. After the growth, the films were slowly cooled down in oxygen atmosphere of ~600 Torr.

$MoS_2$ single crystals were purchased from SPI supplies. The $MoS_2$ flakes were exfoliated using an adhesive tape, which was then pressed against a BTO film and peeled off, leaving the $MoS_2$ flakes on the BTO surface. These flakes were identified by optical microscopy. Due to low optical contrast of thin $MoS_2$ flakes on BTO films, only multi-layer flakes could be found by optical microscopy. The flakes were further characterized by scanning probe microscopy (SPM) and Raman spectroscopy. The thickness of $MoS_2$ flakes used in these studies was in the range from 3 to 8 nm. Raman spectra were recorded using a Thermo Scientific DXR Raman Microscope with a 532 nm excitation laser.

**Scanning Probe Microscopy measurements**. Scanning probe microscopy measurements were conducted by a commercial AFM (MFP-3D-BIO, Asylum Research, USA). Ferroelectric polarization observation and manipulation was achieved using Dual-AC Resonance Tracking Piezoresponse Force Microscopy (DART-PFM)[38]. Conductive-AFM and Kelvin Probe Force Microscopy were used for tunneling current and surface potential characterization, respectively. Silicon AFM probes with Pt/Ir conductive coating and nominal stiffness of 3 N m$^{-1}$ (PPP-EFM, NANOSENSORS) were used to perform the KPFM and PFM measurements. Diamond-coated probes were used for C-AFM measurement for better electrical contact and wear resistance.

DART-PFM imaging was performed by applying an AC modulation bias with amplitude of 0.3–0.5 V$_{peak}$ near the contact resonance frequency that is in the range of 300–350 kHz. Local PFM hysteresis loops were measured at fixed locations as a function of an incrementally changing DC switching bias that superimposed with an AC modulation bias. The DC bias was supplied through the conductive AFM tip, and was off during the measurement of the remanent piezoresponse hysteresis loops. KPFM images were acquired using two-pass technique. The 1st pass is to acquire the morphology profile by tapping mode. During the 2nd pass, AFM tip was lifted and maintained at a constant separation of about 30 nm with respect to the sample surface, while applied an AC bias of 2 V$_{peak}$ to acquire the surface potential images. The scan rate is 1 Hz for both PFM and KPFM images of 256 × 256 pixels.

Optical illumination of sample was realized by using an inverted optical microscope integrated with the AFM system. A white light source is provided by SOLA light engine from Lumencor® USA. A single-band filter set (DAPI-5060C-OMF, Semrock, USA) was installed with over 90% transmission of UV light (377 nm dominated and 50 nm guaranteed minimum bandwidth). The light intensity measured at the sample surface for 40%, 20%, 10%, and 5% UV light intensity is about 5.0, 3.0, 1.9, and 0.7 mW cm$^{-2}$, respectively.

**Modeling**. Calculations of the ground state of triple-layer of $MoS_2$ have been performed with the density functional theory as implemented in the Quantum-Espresso code[39]. Calculations of electronically excited states have been performed with the methods of many-body perturbation theory ("GW approximation" and "Bethe-Salpeter equation") as implemented in the code yambo[34]. The influence of the charge accumulation on the substrate is simulated by a homogeneous electric field (saw-tooth potential) perpendicular to the layer. The Bethe–Salpeter equation is solved on a 30 × 30 two-dimensional k-point grid, ensuring proper convergence of the low-energy excitons and giving the proper energetic ordering of inter- and intra-layer excitons. More details about calculations are given in the Supplementary. We have tested that the same kind of inter-layer excitons exist in bi-layers of $MoS_2$. From the underlying physics (Stark effect, leading to a staggered band gap between the layers at different potential energy), it is clear that dipolar inter-layer excitons with energies below the ones of intra-layer excitons also exist when $n$-layer systems with $n > 3$ are exposed to a perpendicular $E$-field.

**Data availability**. The data that support the findings of this study are available from the corresponding author upon reasonable request.

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

## Acknowledgements

This work was supported by the National Science Foundation (NSF) through Materials Research Science and Engineering Center (MRSEC) under Grant DMR-1420645 (tunnel junction fabrication) and under Grant ECCS-1509874 (electrical characterization). We also acknowledge the support by the Center for Nanoferroic Devices (CNFD), a Semiconductor Research Corporation Nanoelectronics Research Initiative (SRC-NRI) under Task ID 2398.002, sponsored by NIST and the Nanoelectronics Research Corporation (NERC). The work at University of Wisconsin-Madison (thin film fabrication and structural characterization) was supported by the US Department of Energy (DOE), Office of Science, Office of Basic Energy Sciences (BES), under award number DE-FG02-06ER46327. We also acknowledge support from the National Research Fund (FNR), Luxembourg, through projects INTER/ANR/13/20/NANOTMD (E.T. and L.W.), INTER/MOBILITY/16/11467860 2D-Ferro (A.G. and J.I.) and P12/4853155/Kreisel COFERMAT (J.I.).

## Author contributions

T.L. and H.L. conducted electrical and light-induced polarization switching measurements. H.-W.L., J.-W.L., and C.B.E. fabricated the BaTiO$_3$ films. A.L and A.S. prepared and deposited the MoS$_2$ flakes on BaTiO$_3$ for all measurements. J.I., E.T., and L.W. analyzed the MoS$_2$ role in the switching and performed first-principles calculations. A.G. designed and supervised the experimental measurements. T.L., J.I., and A.G. led the discussions and wrote the manuscript. All authors contributed to and commented on the manuscript.

## Additional information

**Competing interests:** The authors declare no competing interests.

