## [Peer Review File · Nature Communications]

Reviewer #1 (Remarks to the Author):

The manuscript of “Optical Control of Polarization in Ferroelectric Heterostructures” Tao Li et al. reported on optically switched polarization in MoS₂/BTO/STO heterostructures. Authors thoroughly carried out their experiment. Optical switching behavior after illumination is very interesting. Experimental results are consistent with their claims. In addition, computational data also support authors' hypothesis. This manuscript can be considered for the publication in Nature Communication after addressing some comments.

1. Why is that the downward polarization in MoS₂/BTO/STO heterojunction is preferred? MoS₂ is n-type semiconductor in which dominant carrier is electron, indicating that the upward polarization might be preferred orientation in above heterostructures. Please clarify it.
2. WSe₂ is p-type dominant materials and has similar optical absorption spectra to MoS₂. If WSe₂ is used instead of MoS₂, optical switching behavior would be reversed. In addition, authors mentioned that bi-directional optical switching is the important challenge. MoTe₂ is ambipolar semiconductor which can be the promising candidate to realize bi-directional optical switching. The reviewer recommend that authors carry out the further experiment using other TMDs such as MoS₂ and MoTe₂, and show preliminary results.
3. Authors have used only ultraviolet (UV) light source. Visible lights over bandgap energy of MoS₂ can also be effective to switch the polarization. Is there any preliminary result under visible illumination? Optical switching behaviors in MoS₂/BTO/STO heterostructures may depend on optical energy (photon) as well as optical power.

Reviewer #2 (Remarks to the Author):

The authors reported a light-induced switching effect in MoS₂/BaTiO₃/SrRuO₃ heterojunction, and attributed it to the interplay between the light-modified screening at the MoS₂/BaTiO₃ interface and the polarization asymmetry of the junction. They also showed that the polarization switching is accompanied with electroresistance switching, which is typical in asymmetric ferroelectric tunnel junctions, and they termed the sizable change of conductance due to light excitation as the optical electroresistance (OER) effect. I think the experimental aspect of this work on revealing the optical switching of ferroelectric polarization is good. To the best of my knowledge, this is the first time to

show such a deterministic switching by optical excitation, despite there were some works focusing on the interplay between light and ferroelectric domain in the literature. However, at present form the paper still needs significant improvement. More results and discussion should be included. The followings are my concerned points that need to be adequately addressed before consideration of acceptance.

Major remarks:

1. The reported optical switching effect seems not very attractive for application, considering the long switching time and obvious degradation. Besides remote control, what are the advantages of such optical switching? I suggest in the introduction section or at the end of discussion the authors compare the optical switching with conventional electrical switching, and also the more recently discussed, mechanical switching.

2. As for the mechanical switching, I would like to mention a recent work [Chen et al., *Journal of the Mechanics and Physics of Solids*, 111, 43–66 (2018)]. This work focused on the effects of surface on mechanical switching of ferroelectric domains, and emphasized that mechanical switching can be obtained based on the interplay between mechanical excitation and the polarization asymmetry of ferroelectric thin film. Such idea is quite relevant to that of present work.

3. Thickness effect needs to be explored. While the authors said that they have investigated BTO films with the thickness ranging from 6 to 12 unit cells, no results of thickness effect on optical switching were shown. The exact thickness of MoS₂ layer in experiment is also unknown. How do the thicknesses of BTO film and MoS₂ layer influence the optical switching behavior? How does the DFT result depend on the thickness of MoS₂ layer?

4. Geometry of the experimental setup shown in Figure 2 is strange. I suppose that the light source is below the heterojunction. How can the light go through the opaque substrate and reach the top surface?

5. Also for the light source, the authors used white light source at the beginning and afterwards UV light was used. Why make such change of light source should be explained.

6. The theoretical aspect of the paper can be improved, on the one hand, by adding more detailed illustration on the DFT result shown in Figure 5, and on the other hand, by showing the polarization asymmetry in MoS₂/BaTiO₃/SrRuO₃ heterojunction via DFT calculation. Screening charge and built-

in field information can be also obtained. Such information is important to verify the mechanism that the authors suggested.

7. According to the mechanism provided by the authors, the field in the MoS₂ layer depends on the polarization direction. Can the downward polarization be also switched into upward polarization, if the light intensity is large enough? Also noteworthy is that during the polarization switching, the field in the MoS₂ layer would decrease and change its sign, the inter-layer exciton can be inverted. This means that the inter-layer exciton can change its role during the switching, and it is the built-in field that plays the dominant role. In other words, the optical switching requires the system has a strong polarization asymmetry. Such an asymmetry hampers bi-directional switching as pointed out at end by the authors. The authors should comment about this.

8. After light illumination, the downward polarization seems to have a smaller PFM amplitude than that of upward polarization as shown in Figure 2c and e. This is in contrast with the case of electrical switching shown in Figure 1. It seems that there is a long characteristic time for the optical process occurring in the MoS₂. I guess PFM test should have different signals at different moments after light illumination (not during illumination)? I also note that after part of the MoS₂ was removed, the exposed BTO surface show much stronger amplitude signal than that of the region still covered by MoS₂ and also that of region of bare BTO surface. Discussion on these abnormal signals would be important to readers who concern experiment details.

Minor remarks:

1. It is not precise to say that MoS₂ is a transition metal dichalcogenide semiconductor with a direct optical gap of ~ 1.9 eV.
2. In the caption of Figure 5, the authors said that the direction of the electric field is shown with a vertical red arrow, but there is actually a black arrow.

Reviewer #3 (Remarks to the Author):

The manuscript with the title of "Optical Control of Polarization in Ferroelectric Heterostructures" claims the optically-driven ferroelectric switching observed in the MoS₂/BTO/SRO system. The most important experimental results are presented in Figure 2 (d), demonstrating the ferroelectric

switching only due to the light illumination. The contents are concise and well-organized. However, the reviewer has doubts about the mechanism of this phenomenon, mainly because the mechanism that the authors address is not limited in this specific MoS₂/BTO/SRO system. To distinguish the contributions of MoS₂ from that of BTO in response of light illumination, the extra experiments such as the change of light wavelength (currently only UV was used; to check the effect of MoS₂, VIS is sufficient not to make the photo-induced charge carriers in BTO), or the change of the 2D materials from n-type MoS₂ to p-type other 2D materials can be considered. (Another mechanical exfoliation experiments for sample preparation might be sufficient.) Currently authors studied this unusual observations only in the framework of MoS₂/BTO/SRO system, the clear analysis on the data cannot be achieved. Also the n-type property of MoS₂ doesn't seem to be considered, which may play an important role of this phenomena.

We would like to thank the reviewers for their valuable comments and constructive criticism. Below, we address these comments point-by-point.

Reviewer #1

Comment 1. Why is that the downward polarization in MoS₂/BTO/STO heterojunction is preferred? MoS₂ is n-type semiconductor in which dominant carrier is electron, indicating that the upward polarization might be preferred orientation in above heterostructures. Please clarify it.

Response. Indeed, there are a number of theoretical and experimental papers reporting that MoS₂ is intrinsically n-type semiconductor. However, there are also many reports showing that in ambient MoS₂ can behave as a p-type semiconductor due to oxygen incorporation,¹ electron traps from adsorbates,² charge transfer from adsorbed water and oxygen³ or contact conditions.^{4,5} Our previous works on the MoS₂/BTO ferroelectric tunnel junctions⁶ and the MoS₂/PZT field effect transistors⁷ also demonstrate that MoS₂ behaves as a p-type semiconductor. We assume that this behavior might be due to the formation of molybdenum oxide at the MoS₂/BTO interface caused by application of the electric potential between MoS₂ and gate electrode. It should be noted that on polymer ferroelectrics (no oxygen) and silicon oxide (with strong Si-O bonds), MoS₂ acts as n-type semiconductor. The p-type conductivity of MoS₂ observed in our studies in combination with the conducting bottom SrRuO₃ electrode, which provides efficient screening for the positive polarization charges, creates conditions favoring the downward polarization.

Comment 2. WSe₂ is p-type dominant materials and has similar optical absorption spectra to MoS₂. If WSe₂ is used instead of MoS₂, optical switching behavior would be reversed. In addition, authors mentioned that bi-directional optical switching is the important challenge. MoTe₂ is ambipolar semiconductor, which can be the promising candidate to realize bi-directional optical switching. The reviewer recommend that authors carry out the further experiment using other TMDs such as MoS₂ and MoTe₂, and show preliminary results.

Response. Following the reviewer's suggestion, we have conducted additional experiment using WSe₂ - a p-type semiconductor, which remains as such even when exposed to air.⁸ Thus, MoS₂ and WSe₂ should behave in the same way in ambient air. Results are shown in Figure R1. By shining the UV light with the same intensity and wavelength on the WSe₂/BTO/SRO junction, we find that the initially upward polarized flake 3 shows the downward polarization after UV illumination, while there is no change in the downward polarization of flakes 1 and 2 due to UV illumination. Thus, the

observed behavior of the $\text{WSe}_2/\text{BTO}/\text{SRO}$ junctions, which employ the p-type 2D semiconductor, is consistent with that of the $\text{MoS}_2/\text{BTO}/\text{SRO}$ junctions in that (i) the optical switching occurs from the upward to the downward direction, and that (ii) MoS_2 behaves as a p-type semiconductor. Experiments involving ambipolar 2D semiconductors, such as MoTe_2 , have been proposed at the end of our manuscript. We are currently working on this experiment and the results will constitute a separate paper. The data in Figure R1 are added to the Supplementary Information as Figure S6 and added this information to the main text on page 11.

Figure R1. UV light induced polarization switching of $\text{WSe}_2/\text{BTO}(12 \text{ u.c.})$ heterostructure. Thickness of WSe_2 is about 1 nm.

Comment 3. Authors have used only ultraviolet (UV) light source. Visible lights over bandgap energy of MoS_2 can also be effective to switch the polarization. Is there any preliminary result under visible illumination? Optical switching behaviors in $\text{MoS}_2/\text{BTO}/\text{STO}$ heterostructures may depend on optical energy (photon) as well as optical power.

Response. We have tried to induce polarization switching using green light (wavelength dominated at 562 nm, which corresponds to photon energy 2.21 eV) is well above the bandgap of multilayer MoS_2 (1.6 eV). Within the same period of time, the green light did not induce any polarization switching as shown in Figure R2. It turned out that UV light was the most effective for optically-induced polarization switching, and we thus focused

our experimental efforts on using UV light. This part of information was included in page 5.

Figure R2. PFM images of MoS₂/BTO (12 u.c.) heterostructure under green light illumination. MoS₂ thickness is approximately 6 nm. The flake was electrically poled to the P_{up} state in dark by applying -6V, 0.1s pulse. The concentric squares created on bare BTO serve as a reference for the polarization orientation. Yellow color in the PFM phase images indicates downward domains, while purple color indicates upward domains, which were generated by scanning with the tip under a dc +/-4V bias. The green light did not switch the polarization from upward to downward direction even after 2.5 hours of light exposure. In Figures R2(g-i), the region where MoS₂ flake was partially removed after illumination is marked by a red dotted triangle. It can be seen that the BTO polarization underneath removed MoS₂ was not switched by green light.

Reviewer #2

Comment 1. The reported optical switching effect seems not very attractive for application, considering the long switching time and obvious degradation. Besides

remote control, what are the advantages of such optical switching? I suggest in the introduction section or at the end of discussion the authors compare the optical switching with conventional electrical switching, and also the more recently discussed, mechanical switching.

Response. This work is mainly a proof-of-concept that the optical illumination can induce ferroelectric polarization switching. As such, our work focuses on the fundamental understanding of the underlying mechanism. As for real device applications, many improvements definitely need to be done especially regarding the switching time. Still, this effect shows a great potential of the 2D semiconductor-ferroelectric structures for future nanoelectronic devices. For example, a possibility of remote control and no need for contacts or applied currents are among the main advantages of the photoinduced switching. The mere activation by light can be an advantage, for applications where the key point is precisely reaction to light (e.g., in smart windows). The sensor need may be avoided, and the switching could be self-powered. We added a short sentence to the conclusion section and on page 3 regarding the rate of optical switching to the conventional electrical one.

Comment 2. As for the mechanical switching, I would like to mention a recent work [Chen et al., *Journal of the Mechanics and Physics of Solids*, 111, 43–66 (2018)]. This work focused on the effects of surface on mechanical switching of ferroelectric domains, and emphasized that mechanical switching can be obtained based on the interplay between mechanical excitation and the polarization asymmetry of ferroelectric thin film. Such ideal is quite relevant to that of present work.

Response. We have included this paper in our reference list.

Comment 3. Thickness effect needs to be explored. While the authors said that they have investigated BTO films with the thickness ranging from 6 to 12 unit cells, no results of thickness effect on optical switching were shown. The exact thickness of MoS₂ layer in experiment is also unknown. How do the thicknesses of BTO film and MoS₂ layer influence the optical switching behavior? How does the DFT result depend on the thickness of MoS₂ layer?

Response. In this work, we employed mechanical-exfoliated multi-layer MoS₂ flakes with thickness of about 3 to 8 nm. The optically-induced polarization switching was found to be very similar in all samples irrespective of the MoS₂ and BTO thicknesses in these ranges. We added information on the MoS₂ thickness to the text (page 20).

Note that it would be increasingly difficult to photo-switch thicker BTO films, as we are dealing with an activated process that will be more effective in thinner films. If we think

only of thermodynamic equilibrium arguments, a perfect (defect-free) thicker BTO film should be in principle photo-switchable as well. It is true that the thicker MoS₂, the more effectively the field caused by the interface charge σ will be screened away from the interface. However, as long as the light can be absorbed close to the MoS₂/BTO interface, the thickness for multilayer MoS₂ flakes should not make significant differences.

Concerning the DFT results for the absorption spectrum of the MoS₂ layer, our calculations show that the differences between having two or three layers are not significant, and hence we can safely assume our conclusions will remain the same for thicker MoS₂ films. (Note we did find that the result for a single MoS₂ layer is qualitatively different, but this one is not important in the present discussion.)

Comment 4. Geometry of the experimental setup shown in Figure 2 is strange. I suppose that the light source is below the heterojunction. How can the light go through the opaque substrate and reach the top surface?

Response. Our AFM system is built on top of an inverted optical microscope. The light path is through a UV filter and an inverted optical lens. In order to precisely control the wavelength, light intensity and illuminated location on sample, shining from the bottom of the heterostructure is the limit for our current system. However, the STO substrate is not opaque but rather semitransparent. The light intensity measured after the light passes through BTO/SRO/STO structure is about 5 mW/cm² as described on page 5, which is sufficient for the optically induced polarization switching. It means that only moderate light power is required for such switching. Additionally, it implies that if light intensity can be increased and directly shine from the top of the heterostructure, a much faster switching speed can be expected.

Comment 5. Also for the light source, the authors used white light source at the beginning and afterwards UV light was used. Why make such change of light source should be explained.

Response. The preliminary tests have been performed using a white light source (SOLA light engine) to identify the overall optical effect on TER in the FTJ. The white light spectrum of that source ranges from red to UV, including the UV wavelength that we used. Later, to clarify the mechanism of optically induced polarization switching, monochromatic light was used, and we found that only UV light switched the polarization (see our Response to comment 3 by Reviewer #1).

Comment 6. The theoretical aspect of the paper can be improved, on the one hand, by adding more detailed illustration on the DFT result shown in Figure 5, and on the other hand, by showing the polarization asymmetry in MoS₂/BaTiO₃/SrRuO₃ heterojunction via DFT calculation. Screening charge and built-in field information can be also obtained. Such information is important to verify the mechanism that the authors suggested.

Response. In the previous version of the manuscript, all detailed information about the calculations were “hidden” in the Supplementary Information. We added a short description (with a reference to the codes that we have used) in the “Materials and Methods” section of the main manuscript. We also edited the detailed description of the calculation and its results in the Supplementary Information. Furthermore, we added an additional panel to Figure 5, showing a zoom into the band-structure, focusing on the valence-band maximum and conduction band minimum at K, displaying all the features of a “staggered” (type II) direct gap. We also gave reference to a review article on optical properties and excitonic effects in few-layer systems of MoS₂, which contains essential information to understand the present calculations.

Regarding the explicit simulation of the MoS₂/BaTiO₃/SrRuO₃ junction, we agree with the referee that it would be very interesting and fruitful to tackle it. Nevertheless, we believe it falls beyond the scope the present manuscript, for several reasons. On the one hand, from a practical perspective, we should note this would be computationally very challenging, as it would require very accurate spin-polarized calculations (energy resolution better than 1 meV/atom is required to treat BTO ferroelectric phase) in a slab of at least 100 atoms that combines metallic and semiconducting parts, and where one should allow for a sufficiently wide vacuum region to avoid effects associated with the slab’s polarity. Those are, from many perspectives, very challenging simulations.

On the other hand, and more importantly, the identification of a realistic model system to simulate is far from trivial and constitutes a difficult research project on its own right, in our opinion. One major issue pertains to the nature of doping in MoS₂. As mentioned above, we have good reasons to believe that our semiconductor film is p-type, so in principle we can build on that information. Yet, to capture this p-character, we would probably need to explicitly include dopants in the simulation, in an appropriate amount, which complicates things greatly as (1) the specific choice of dopants might influence the results, and (2) we might need to consider simulation boxes significantly larger than the minimal one for the ideal, un-doped cases. (Hence, instead of the 100 atoms mentioned above, the simulation could easily grow to about 400, at least.) Let us stress that usual ‘short-cuts’ to treat doping, like controlling by-hand of the number of electrons in the simulation and adding a compensating background charge, will not work in a complex heterostructure in a slab geometry like this one.

Even if we were able to treat MoS₂ in a realistic way, we would still face a second major difficulty: the atomistic model of the BTO/MoS₂ interface. (We can assume that the

BTO/SRO interface is comparatively simple.) Note that the details of this interface may strongly influence our quantitative results for the built-in field inside the BTO junction. Then, unfortunately, predicting a realistic interface structure is far from trivial, even if we assume a ‘clean’ frontier between the two materials. In our minds, this is somewhat reminiscent of the problem of the interface between perovskite oxides and materials like Si or SiO₂, which has occupied researchers for years.

For these reasons, an explicit and realistic (thus meaningful) first-principles study of the MoS₂/BTO/SRO junction is a challenging problem that requires a dedicated mid-term effort. Hence, in this work we rely as much as possible on the known experimental facts, which we take as a starting point for our theoretical discussion. We realize this is not the ideal first-principles philosophy and we hope to address these lingering issues in the forthcoming work. Nevertheless, we do believe that the present one is a well-justified approach that has proved very useful to rationalize our experimental results.

Comment 7. According to the mechanism provided by the authors, the field in the MoS₂ layer depends on the polarization direction. Can the downward polarization be also switched into upward polarization, if the light intensity is large enough? Also noteworthy is that during the polarization switching, the field in the MoS₂ layer would decrease and change its sign, the inter-layer exciton can be inversed. This means that the inter-layer exciton can change its role during the switching, and it is the built-in field that plays the dominant role. In other words, the optical switching requires the system has a strong polarization asymmetry. Such an asymmetry hampers bi-directional switching as pointed out at end by the authors. The authors should comment about this.

Response. The light intensity only affects the switching speed, but not the switching direction. The direction of optically induced polarization switching primarily depends on the the direction of the build-in field in BTO which, in turn, depends on the availability of mobile carriers in SrRuO₃ and MoS₂, as well as on the details of their interfaces with BTO. In the case of the MoS₂/BTO/SRO, switching can proceed only from the upward to the downward direction.

Note that the difficulty about bi-directional switching with light in our MoS₂/BTO/SRO system is the following: how to use light (instead of a bias) to accumulate negative carriers at the MoS₂/BTO interface, so that we obtain $\sigma < 0$ and develop the E_σ field inside BTO that (1) counteracts the built-in field E_{bi} and forces P_{down} to switch to P_{up} . The problem is that, in the P_{down} state, σ will be approximately 0 and no large fields will exist inside MoS₂. In such conditions, light will not generate any inter-layer excitons in MoS₂, and we will not induce accumulation of charge (positive or negative) at the interface.

Indeed, according to our picture (and to our experimental evidence), in the P_{down} state illumination does not result in a charge rearrangement in the MoS₂ flake.

Hence, according to both our experimental observations and proposed physical picture, illumination alone does not allow us to destabilize the P_{down} state of our MoS₂/BTO/SRO junction. Further, as the referee concludes, strong polarization asymmetry is important to explain the observed switching. We have revised our explanation on page 11 to clarify these issues.

Comment 8. After light illumination, the downward polarization seems to have a smaller PFM amplitude than that of upward polarization as shown in Figure 2c and e. This is in contrast with the case of electrical switching shown in Figure 1. It seems that there is a long characteristic time for the optical process occurring in the MoS₂. I guess PFM test should have different signals at different moments after light illumination (not during illumination)? I also note that after part of the MoS₂ was removed, the exposed BTO surface show much stronger amplitude signal than that of the region still covered by MoS₂ and also that of region of bare BTO surface. Discussion on these abnormal signals would be important to readers who concern experiment details.

Response. Electrically-induced polarization switching through MoS₂ unavoidably involves certain charge injection, which may also alter the electronic structure of MoS₂ locally.^{9,10} In contrast, the light-induced polarization switching relies on the available number of the photocarriers without any substantial charge injected from external source. Note also that PFM amplitude for optically switched structure depends on the illumination time (see Figures 2(f-k)). This can be the reason of amplitude discrepancy between the PFM images of the electrically and optically switched structures. Furthermore, the PFM amplitude is affected not only by the polarization states, but also by the surface charges. Figure R3 shows electrically poled bare BTO film, which has higher amplitude than the as-grown BTO surface. Even the upward and downward domains created by electrical scanning of the same magnitude of bias show different PFM amplitude values. It is either due to that the non-poled region is not completely polarized in a single direction, or the screening conditions by the surface charges are different for poled regions. Removal of the MoS₂ flake after optical switching, will expose the non-screened polarization charges affecting the PFM amplitude signal. This explains the different amplitude signals for BTO region under MoS₂ and those for bare BTO. The interplay between electromechanical and electrostatic contributions in PFM has been discussed in details in recent reviews.

Figure R3. Electrically-poled bare BTO with upward and downward polarization states showing different amplitude strength from different domains.

Comment 9. It is not precise to say that MoS₂ is a transition metal dichalcogenide semiconductor with a direct optical gap of ~1.9 eV.

Response. The statement has been revised (page 3).

Comment 10. In the caption of Figure 5, the authors said that the direction of the electric field is shown with a vertical red arrow, but there is actually a black arrow.

Response. We thank the reviewer for pointing this out. We corrected this in the text.

Reviewer #3

Comment 1. The manuscript with the title of "Optical Control of Polarization in Ferroelectric Heterostructures" claims the optically-driven ferroelectric switching observed in the MoS₂/BTO/SRO system. The most important experimental results are presented in Figure 2 (d), demonstrating the ferroelectric switching only due to the light illumination. The contents are concise and well-organized. However, the reviewer has doubts about the mechanism of this phenomenon, mainly because the mechanism that the authors address is not limited in this specific MoS₂/BTO/SRO system. To distinguish the contributions of MoS₂ from that of BTO in response of light illumination, the extra experiments such as the change of light wavelength (currently only UV was used; to check the effect of MoS₂, VIS is sufficient not to make the photo-induced charge carriers in BTO), or the change of the 2D materials from n-type MoS₂ to p-type other 2D materials can be considered. (Another mechanical exfoliation experiments for sample preparation might be sufficient.) Currently authors studied this unusual observations only in the framework of MoS₂/BTO/SRO system, the clear analysis on the

data cannot be achieved. Also the n-type property of MoS₂ doesn't seem to be considered, which may play an important role of this phenomena.

Response. These comments are relevant to the comments 1, 2 and 3 by Reviewer #1, which we addressed above. Results shown in Figures R1 and R2 also directly address the reviewer's comments regarding the effect of the wavelength and the use of another 2D material. We hope that the reviewer will find our responses satisfactory.

1. Neal AT, Pachter R, Mou S. P-type conduction in two-dimensional MoS₂ via oxygen incorporation. *Appl Phys Lett* **110**, 193103 (2017).
2. Woanseo P, *et al.* Oxygen environmental and passivation effects on molybdenum disulfide field effect transistors. *Nanotechnology* **24**, 095202 (2013).
3. Tongay S, *et al.* Broad-range modulation of light emission in two-dimensional semiconductors by molecular physisorption gating. *Nano Lett* **13**, 2831-2836 (2013).
4. Chuang S, *et al.* MoS₂ P-type transistors and diodes enabled by high work function MoO_x contacts. *Nano Lett* **14**, 1337-1342 (2014).
5. Dolui K, Rungger I, Sanvito S. Origin of the n-type and p-type conductivity of MoS₂ monolayers on a SiO₂ substrate. *Phys Rev B* **87**, (2013).
6. Li T, *et al.* Polarization-mediated modulation of electronic and transport properties of hybrid MoS₂-BaTiO₃-SrRuO₃ tunnel junctions. *Nano Lett* **17**, 922-927 (2017).
7. Lipatov A, Sharma P, Gruverman A, Sinitskii A. Optoelectrical molybdenum disulfide (MoS₂)-ferroelectric memories. *ACS nano* **9**, 8089-8098 (2015).
8. Wang S, Zhao W, Giustiniano F, Eda G. Effect of oxygen and ozone on p-type doping of ultra-thin WSe₂ and MoSe₂ field effect transistors. *Phys Chem Chem Phys* **18**, 4304-4309 (2016).
9. Liu Q, Li L, Li Y, Gao Z, Chen Z, Lu J. Tuning electronic structure of bilayer MoS₂ by vertical electric field: a first-principles investigation. *J Phys Chem C* **116**, 21556-21562 (2012).
10. Santos EJG, Kaxiras E. Electrically driven tuning of the dielectric constant in MoS₂ layers. *ACS nano* **7**, 10741-10746 (2013).

Reviewer #1 (Remarks to the Author):

Authors have tried to address all of comments. Authors also carried out additional experiments to clarify points raised by the reviewer. The revised manuscript is worth publishing Nature Communication. The reviewer put on a minor comments. For optical switching, blue light may be effective. The reviewer thought that there is threshold photon energy for optically-induced polarization.

Reviewer #2 (Remarks to the Author):

The authors have adequately addressed the points that the reviewer raised. The manuscript is now suitable for publishing on Nature Communications.

Reviewer #3 (Remarks to the Author):

Even though the authors have presented the answers to the questions that the reviewer raised, several important issues have not been clarified yet.

1) The light-induced polarization switching direction : Authors argued that their MoS₂ flake is p-type unlike the normal status of MoS₂. For the clear demonstration of the optically-induced ferroelectric switching, the case of standard MoS₂ flake, in other words, the case of n-type 2D flake should be presented. This is the issue whether the ferroelectric polarization can be compensated by 2D layer or the surface charges irrespective of the 2D layer. I think that the present work is corresponding to the latter case, that is the surface charge issue which is not related to the type of 2D materials on top of BTO. Under this assumption, the role of 2D material is the absorption of light to generate large number of electrons irrespective of the initial carrier type of materials. The abundant electrons in the top layer may influence the polarization switching behavior. Phenomenologically this observation of photo-induced ferroelectric switching behavior is very interesting, but it doesn't seem to be controllable even in the basic polarization direction in the current stage.

2) This is the minor concern, but I have a question whether the UV light (377 nm) is transparent for BTO. In the quick web-site search, many absorption curves of BTO as a function of wavelength seem

to show some absorption features near 377 nm. If there is certain amount of photo-induced carrier generation in the BTO, they might be coupled with MoS₂ on top of the BTO, which makes the switching easier I guess. If every experiments were done with the green light that is known to be transparent to BTO, there would be no doubt for that.

In summary, the possible coupling of i) the photo-electrons generated in BTO and that of ii) the surface charges (or interfacially trapped charges) between MoS₂ and BTO with the photo-induced MoS₂ carrier may play a role in the process of ferroelectric switching. The resubmitted manuscripts and the rebuttals did not include the proper answers to the above questions.

Reviewer #1

Comment. Authors have tried to address all of comments. Authors also carried out additional experiments to clarify points raised by the reviewer. The revised manuscript is worth publishing Nature Communication. The reviewer put on a minor comments. For optical switching, blue light may be effective. The reviewer thought that there is threshold photon energy for optically-induced polarization.

Response. We appreciate a positive conclusion of the reviewer regarding acceptability of the manuscript. We would like to add that further experiments are under way to investigate in more detail the wavelength effect on the optical switchability of the MoS₂/BTO tunnel junctions.

Reviewer #2

Comment. The authors have adequately addressed the points that the reviewer raised. The manuscript is now suitable for publishing on Nature Communications.

Response. We appreciate a positive conclusion of the reviewer regarding acceptability of the manuscript.

Reviewer #3

Comment 1. The light-induced polarization switching direction : Authors argued that their MoS₂ flake is p-type unlike the normal status of MoS₂. For the clear demonstration of the optically-induced ferroelectric switching, the case of standard MoS₂ flake, in other words, the case of n-type 2D flake should be presented. This is the issue whether the ferroelectric polarization can be compensated by 2D layer or the surface charges irrespective of the 2D layer. I think that the present work is corresponding to the latter case, that is the surface charge issue which is not related to the type of 2D materials on top of BTO. Under this assumption, the role of 2D material is the absorption of light to generate large number of electrons irrespective of the initial carrier type of materials. The abundant electrons in the top layer may influence the polarization switching behavior. Phenomenologically this observation of photo-induced ferroelectric switching behavior is very interesting, but it doesn't seem to be controllable even in the basic polarization direction in the current stage.

Response. We absolutely agree with the reviewer in that the role of 2D material is the absorption of light to generate large number of electrons irrespective of the initial carrier type of materials and this is in line with the physical picture we propose in our paper. We understand and share the reviewer's desire to get a deeper understanding of the observed optically induced polarization switching. It would be indeed great to get

data using n-type 2D semiconductor. The problem is that MoS₂, used in our study, is “the most n-type” material in the TMD family, and it already behaves as a p-type semiconductor on BTO (Fig R1). So, other TMDs are not good candidates to test the effect of the n-type behavior the reviewer asks about. In the revised manuscript we have provided new experimental data on WSe₂, which behaved similarly to MoS₂, and thus other TMDs are expected to show in such devices a p-type behavior as well. Outside of the TMD family, we do not immediately see suitable monolayer n-type semiconductors that could be tested. For example, phosphorene is a p-type semiconductor as well. Besides, the proposed mechanism of the observed effect is based on inter-layer excitons in the TMD layer, so a non-TMD material is *a priori* expected to behave differently and cannot be used either.

We also would like to reiterate, that the reports on MoS₂ behaving as a p-type semiconductor are not very unusual, and in the previous response letter we provided 7 papers demonstrating various examples of this phenomenon. Additionally, we would like to draw your attention to another paper that reports MoS₂ behaving as a p-type material: Capasso et al, Adv. Energy Mat. **6**, 1600920 (2016). In this paper, MoS₂ plays a role of a hole transport layer when combined with perovskite solar cells. This suggests that while pristine MoS₂ is clearly n-type, it can become p-type when it is brought in contact with other materials even in the absence of the optical excitation. We assume that passivation of the BTO layer may allow us to maintain the n-type MoS₂. We are investigating this assumption and hope to publish the results in a separate paper in the future.

Figure R1. Calculated band alignment of TMD monolayers. Reproduced from Kang et al. Appl. Phys. Lett. **102** 012111 (2013).

Let us also mention that from what this reviewer writes it seems that he/she is essentially convinced by our explanation. For example, he/she writes “Under this assumption, the role of 2D material is the absorption of light to generate large number of electrons irrespective of the initial carrier type of materials”, which is perfectly in line with the physical picture we propose in our paper. Similarly, when he/she writes “The

abundant electrons in the top layer may influence the polarization switching behavior. Phenomenologically this observation of photo-induced ferroelectric switching behavior is very interesting”, again we find ourselves in essential agreement with the reviewer. As regards his/her last regret (“but it doesn’t seem to be controllable even in the basic polarization direction in the current stage”), we totally agree that this is the open challenge for the future, as mentioned in the final outlook in our paper.

Comment 2. This is the minor concern, but I have a question whether the UV light (377 nm) is transparent for BTO. In the quick web-site search, many absorption curves of BTO as a function of wavelength seem to show some absorption features near 377 nm. If there is certain amount of photo-induced carrier generation in the BTO, they might be coupled with MoS₂ on top of the BTO, which makes the switching easier I guess. If every experiments were done with the green light that is known to be transparent to BTO, there would be no doubt for that.

Response. The 377 nm light is at the edge of the BTO absorption spectrum. However, most of the earlier studies of the photoinduced phenomena in BTO were done for bulk crystals and films that were much thicker than those used in the present work. Nevertheless, even if we consider this mechanism, light absorption and generation of excess electrons in BTO would lead the more efficient screening of the upward polarization, which means that polarization will become more stable and no switching should be observed (as is demonstrated by our experiment using bare BTO films). It is the presence of the MoS₂ layer that is necessary to induce polarization reversal, as we show experimentally. And it is the generation and separation of the photocarriers in the MoS₂ layer that can destabilize the upward polarization leading to its switching as is proposed by our model. In any case, we agree with the reviewer that further studies will be necessary to clarify the wavelength dependence of the observed effect. In our paper, we just unambiguously demonstrated the observed effect in the MoS₂/BTO junctions for the UV light and we are glad that the reviewer finds this finding very interesting.

Comment 3. In summary, the possible coupling of i) the photo-electrons generated in BTO and that of ii) the surface charges (or interfacially trapped charges) between MoS₂ and BTO with the photo-induced MoS₂ carrier may play a role in the process of ferroelectric switching. The resubmitted manuscripts and the rebuttals did not include the proper answers to the above questions.

Response. We would like to note that we did address the earlier comments of the reviewer related to the use of light of different wavelengths and other 2D materials. These new results had been added to the revised version. Also, as we mentioned in our response to this reviewer’s Comment 2, generation of photoelectrons in BTO that would accumulate on the BTO surface in the case of the upward polarization would not by itself lead to polarization reversal. On the contrary, the polarization will be more efficiently screened precluding any switching. Thus, it is a combination of the MoS₂ and BTO layers that is required to observe the photo-induced switching effect (we have

emphasized this again on page 5). In this regard, there is no disagreement between the reviewer's statement and our model that infers an interplay between the induced charges in BTO and MoS₂ as a mechanism behind the photo-induced switching. We also added a note on page 12 that it is necessary to perform careful studies of the wavelength dependence of the optical switching effect to address the role of the carriers in BTO. This could be a subject of a separate paper.

In summary, we believe that we have properly addressed this comment related to the role of the BTO and the MoS₂ layers in the observed effect. Let us also add that we greatly appreciate the fact that the reviewer finds our work valuable.

Reviewer #1 (Remarks to the Author):

Authors have addressed comment. The reviewer does not have any comment. The revised manuscript is worth publishing Nature Communication.

Reviewer #3 (Remarks to the Author):

In the supporting information file of the revised manuscript, the authors added the experimental results of WSe₂/BTO/SRO junction, showing a good polarization switching from UP to DOWN by UV irradiation. Because the conversion of intrinsic n-type MoS₂ to p-type material on the surface of BTO is not the main issue of this manuscript, the reviewer thinks that the additional WSe₂ experiment in the revised manuscript is now sufficient for the demonstration of this optically-driven ferroelectric polarization switching phenomenon. Also the phrase of 'p-type' MoS₂ seems to be eliminated in the main manuscript, which also helps the reader of this article not to be confused about the carrier-type of 2D material employed as an electrode on top of BTO. The reviewer agrees to the publication of this manuscript to Nature Communications.